# Distribution, Characterization and the Commercialization of Elite Rhizobia Strains in Africa

**DOI:** 10.3390/ijms23126599

**Published:** 2022-06-13

**Authors:** Clabe Wekesa, Abdul A. Jalloh, John O. Muoma, Hezekiah Korir, Keziah M. Omenge, John M. Maingi, Alexandra C. U. Furch, Ralf Oelmüller

**Affiliations:** 1Matthias Schleiden Institute of Genetics, Bioinformatics and Molecular Botany and Plant Physiology, Friedrich-Schiller-University Jena, Dornburger Str. 159, 07743 Jena, Germany; clabe.wekesa@uni-jena.de (C.W.); keziah.moraa.omenge@uni-jena.de (K.M.O.); alexandra.furch@uni-jena.de (A.C.U.F.); 2International Centre of Insect Physiology and Ecology, P.O. Box 30772, Nairobi 00100, Kenya; ajalloh@icipe.org; 3Department of Biological Sciences, Masinde Muliro University of Science and Technology, P.O. Box 190, Kakamega 50100, Kenya; jmuoma@mmust.ac.ke; 4Crops, Horticulture and Soils Department, Egerton University, P.O. Box 536, Egerton 20115, Kenya; hezkorir@yahoo.com; 5Department of Biochemistry, Microbiology and Biotechnology, Kenyatta University, P.O. Box 43844, Nairobi 00100, Kenya; maingi.muthini@ku.ac.ke

**Keywords:** elite rhizobia, legumes, biological nitrogen fixation, commercialization of elite rhizobia

## Abstract

Grain legumes play a significant role in smallholder farming systems in Africa because of their contribution to nutrition and income security and their role in fixing nitrogen. Biological Nitrogen Fixation (BNF) serves a critical role in improving soil fertility for legumes. Although much research has been conducted on rhizobia in nitrogen fixation and their contribution to soil fertility, much less is known about the distribution and diversity of the bacteria strains in different areas of the world and which of the strains achieve optimal benefits for the host plants under specific soil and environmental conditions. This paper reviews the distribution, characterization, and commercialization of elite rhizobia strains in Africa.

## 1. Introduction

Nitrogen is a crucial constituent of amino acids (proteins), urea, nucleic acids, nicotinamide adenine dinucleotide, and adenosine triphosphate in all living cells. It is a primary component of photosynthetic pigment (chlorophyll) that harvests light energy essential for plant growth and biomass production [1,2]. Dinitrogen exists as a gas and is the main component of the atmosphere, constituting about 78% of air. However, only the oxidized (nitrates) or reduced (ammonium) forms of nitrogen can be utilized by plants. Dinitrogen can be reduced to ammonium (NH_3_) through the Haber–Bosch process or via biological nitrogen fixation (BNF) utilizing some soil bacteria or archaea (diazotrophs). Rhizobia belong to *Alphaproteobacteria* and *Betaproteobacteria*, a group of Gram-negative bacteria that forms nodules on roots (sometimes stems) of leguminous plants to fix nitrogen in a symbiotic relationship with their host plants [3]. The rhizobia–legume symbiosis is the most studied plant–microbial mutualism [4,5,6] because of the importance of nitrogen fixation for almost all agricultural systems.

The major legumes in Africa are common bean (*Phaseolus vulagaris* L.), soybean (*Glycine max*), pigeon pea (*Cajanus cajan*), broad bean or fava/faba bean (*Vicia faba*), groundnut (*Arachis hypogaea*), chickpea (*C. arietinum*), and cowpea (*Vigna unguiculata* L. Walp.) [7]. Legume cultivation in most African countries is performed by substance farmers primarily for food, although the surplus can be sold as a source of income [8]. Because of the widespread poverty in rural Africa, farmers cannot afford nitrogenous fertilizers, and therefore cultivation is dependent primarily on indigenous rhizobia for provision of nitrogen requirements. However, in some regions, local isolates are not effective in nitrogen fixation, and low legume yields with those isolates were repeatedly reported in East Kenya [9,10], West Africa [11], and South Africa [12]. Commercial rhizobia inoculants have been successful in some regions [13,14]; however, numerous studies also demonstrate that they failed to promote plant growth and final yield in many agricultural areas [15,16,17,18]. Failure of the commercial rhizobia in African farms might be due to poor adaptation of the isolates to the local soil conditions. Most formulations contain strains isolated from other continents [8], which might not be sufficiently adapted to the local conditions. It is, therefore, necessary to isolate and investigate the rhizobial strains in the local soils and utilize them as commercial inoculants in specific regions in Africa. However, the development and commercialization of formulations with local elite isolates from the particular areas require substantial biological and ecological studies before successful application.

First, this requires studies on the symbiotic efficiency of the newly isolated elite rhizobia on the local plant species both in greenhouse and field studies. The isolates’ ability to withstand abiotic stress such as harsh environmental conditions caused by pH extremes, heavy metal toxicity, high temperatures, or osmotic stress should be determined. Furthermore, they must successfully compete with other indigenous rhizobia present in the local rhizosphere. Therefore, nodule occupancy of the new strains in the presence of the indigenous rhizobia in the local soil should be investigated. This review discusses various studies on the distribution, characterization, and commercialization of rhizobia in Africa.

## 2. Nodulation and Nitrogen Fixation Processes

Nodulation is a host-specific process, with each rhizobia having a specific range of hosts [19]. It is a multistep process involving the host plant and its symbionts in the rhizosphere. Under severe nitrogen starvation, legume roots release flavonoid-containing exudates in the rhizosphere, interacting with nodulation protein D (NodD) in rhizobia [20]. Activated NodD induces the expression of *nod* genes through its binding to the *nod* operon. This causes the synthesis and release of nod factors, lipochitooligosaccharide(LCO)-based signaling molecules whose interaction with specific genes in the root hairs leads to cortical cell divisions forming root nodule primordia and simultaneously initiates an infection process to deliver the bacteria into the nodule cells. The infection of most legumes involves developing plant-made infection threads through root hair branching, deformation, and curling [21]. There are two nodule types: indeterminate (e.g., in *Medicago*, pea, and clover) and determinate (e.g., in common beans and soybeans). Indeterminate nodules are thought to originate from cell divisions in the inner cortex and usually possess a persistent apical meristem. They are cylindrical, with a developmental gradient from the apex to the base of the nodule, which may divide into different nodule zones [22]. However, determinate nodules result from cell divisions in the middle/outer cortex of the root, lack a persistent meristem, and are spherical. Cell divisions of a determinate nodule stop at early developmental stages. Therefore, mature nodule develops through cell enlargement; as such, the infected cells develop more or less synchronously to the nitrogen-fixing stage [23].

The infection threads harboring the dividing bacteria grow through an epidermal cell layer into the nodule primordial cells (Figure 1). The bacteria are then released and internalized in an endocytosis-like process by the cortical cells. In nodule cells, individual bacteria are enclosed by a membrane of plant origin, forming an organelle-like structure called the symbiosome. The bacteria further proliferate and differentiate into nitrogen-fixing bacteroids [23] that actively synthesize nitrogenase enzymes. Nitrogenase is extremely sensitive to oxygen; thus, the nodule must maintain low oxygen concentrations while maintaining oxygen supply for bacterial metabolism [24]. Infected cells synthesize leghemoglobin that is thought to buffer free oxygen in the nanomolar range, avoiding the inactivation of oxygen-labile nitrogenase while maintaining high oxygen flux for respiration [25]. Abolishing leghemoglobin synthesis in *Lotus japonicus* caused an increase in nodule free oxygen, a decrease in the ATP/ADP ratio, loss of rhizobial nitrogenase protein, and an absence of nitrogen fixation [26].

The symbiosome membrane acts as the interface between eukaryotic and prokaryotic symbionts, and thus it possesses transporters for nutrient exchange between the symbiotic partners [27]. The host plant provides carbon sources for bacteroid activities in the form of dicarboxylates, malate, and succinate [28]. Phosphoenolpyruvate carboxylase and malate dehydrogenase convert the carbon flux from glycolysis to form malate, which can be taken directly by bacteroids [29]. Ammonia exported from the bacteroids diffuses into the cytosol of the infected host cells, where it is converted to glutamine (Gln) and glutamate (Glu) by the enzymes Gln synthetase and Glu synthase. In indeterminate nodules, Glu and Gln are further converted to aspartate (Asp) and Asn by Asp aminotransferase and Asn synthetase [30]. In indeterminate nodules, Gln further enters the purine synthesis pathway and is converted to ureides (McLaughlin et al., 1987) as the final product exported to the host plant.

In addition to root hair entry and through infection thread observed in most rhizobia, another mechanism known as crack entry occurs in *Bradyrhizobium*. The infection by crack entry necessitates penetration of the bacterial microsymbiont through epidermal breaches located at sites where lateral roots (root epidermis) or adventitious roots (stem epidermis) protrude and so provide penetration sites at the fissure [31,32]. After entry, the cells occupy the space between epidermal and cortical cells. Beneath the axillary root hairs, basal cells become enlarged. Such enlarged basal cells are the first to become infected by some of the invading rhizobia cells, while others continue to spread intercellularly [33]. After intercellular spreading of the rhizobial cells, the cell walls of particular plant cells that will eventually internalize the bacteria are structurally altered and appear partially degraded, as in a cellulolytic process. Ultimately, the plasma membrane is partly exposed to the bacteria in such invaginations, and the bacteria are internalized in host cells in an undefined matrix [33]. The encapsulated cells multiply rapidly within the infected plant cell. In the beginning, they are enclosed in the same membrane envelope, but the envelope is later divided so that each cell becomes enclosed separately [34]. The invaded plant cell also divides rapidly, thereby distributing the endophytes such as other cell organelles. The nodule tissue eventually originates from one or few infected plant cells.

The plant must maintain a balance between providing carbon to bacteroids for nitrogen fixation while retaining enough carbon for growth and development and keeping optimum nitrogen efflux from the nodules. It was demonstrated in common beans that nitrogen from senescing lower leaves re-translocated back to the nodules is involved in negative feedback regulation of nitrogen fixation [35]. A conserved long-distance mechanism called autoregulation of nodulation (AON) enables the host plant to control the number of nodules and, in turn, control energy consumption [36]. The onset of AON is related to the production of the CLE (CLAVATA3/embryo-surrounding region) peptides and root-derived mobile signals associated with nodulation inhibition [37]. The produced CLE peptides are transported to the leaves via the xylem, where they are recognized by specific receptors [38]. This ligand–receptor interaction elicits shoot-derived secondary signals transported down to the roots, where they inhibit further nodulation [39,40].

There are primarily two clusters of nitrogen-fixing genes: nitrogen fixation (*nif*) and fixation (*fix*) genes. Cluster I contains *nif*HDKE, *nif*S, nifN, *fix*ABCX *nif*A *nif*B, and *frd*X genes, while cluster II contains *fix*LJ, *fix*K, *fix*NOQP, *fix*GHIS genes [41]. Nitrogenase is a complex consisting of component I (molybdenum-iron-containing protein) and component II (iron-containing protein). Nitrogen reduction occurs in component I, while component II is an obligate electron donor to component I [42,43]. Component I proteins are encoded by *nif*D and *nif*K genes, while component II protein is encoded by the *nif*H gene [44]. NifBEN proteins are involved in the synthesis of component I MoFe cofactor through uncharacterized mechanisms [45,46]. NifS catalyzes the formation of elemental sulfur by using L-cysteine as a substrate used in the synthesis of nitrogenase metalloclusters [47]. *fdx*N gene, which lies downstream of the *nif*B gene, encodes ferredoxin-like electron transfer proteins, essential for nitrogen fixation [48].

It was demonstrated that the *fix*ABCX gene products are involved in electron transport to nitrogenase [49]. To show the significance of these genes, a mutation on any of the genes abolished nitrogen fixation in *R. meliloti* [50]. *fix*NOQP gene products are involved in the synthesis of a membrane-bound, cytochrome C-containing heme/copper cytochrome oxidase [51] required to support bacteroid respiration under conditions of low oxygen present in the root nodules [51]. *fix*GHIS genes located downstream of the *fix*NOQP gene encode for transmembrane proteins essential in producing a high-affinity terminal oxidase necessary for bacterial respiration in the microaerobic nodule [52].

Nitrogen fixation is an energy-intensive process that requires strict control, e.g., it requires 40 moles of ATP to reduce a single molecule of dinitrogen to ammonia [53,54]. In rhizobia, the regulation of nitrogen fixation is mainly controlled by the amount of free oxygen in the nodule, unlike free-living bacteria, whose nitrogen fixation is regulated by the amount of fixed nitrogen in the rhizosphere [41,55]. NifA and FixK constitute two central cascades that regulate nitrogen fixation genes [56]. NifA targets nitrogenase complex encoding and accessory genes (*nif*HDKE, *nif*N, *fix*ABCX, *nif*A, *nif*B, and *frd*X), while FixK is involved in the transcriptional regulation of *fix*NOQP and *fix*GHIS genes [41]. Figure 2 below is the current model that depicts the regulation of nitrogen fixation rhizobia.

## 3. Distribution of Rhizobia in African Soil

Leguminosae comprises more than 19,000 species [57] that have independently evolved in different places. For example, *Phaseolus* was developed in Andean and Mesoamerican regions [58,59], *Medicago*, *Trifolium*, and *Pisum* in the Middle East [60], soybean in Asia [61], *Galega oficinalis and G. orientalis* in the Caucasus [62], and Bambara groundnuts as well as cowpea in Africa [63]. Several older legumes, such as acacia, have evolved world-wide [64]. However, legumes are currently distributed throughout the world. The principal symbionts of particular legumes seem to have developed locally, coinciding with the legumes’ distribution (Table 1) [65].

**Table 1 ijms-23-06599-t001:** Rhizobia isolated from African soils.

Strain	Legume	Reference
*R. phaseoli*	*P. vulagaris*	[17,66]
*R. paranaense*,	*P. vulagaris*	[66]
*R. sophoriradicis*	*P. vulagaris*	[66]
*R. leucaenae*	*P. vulagaris*	[66]
*R. aegyptiacum*	*P. vulagaris*	[66]
*R. tropicii*	*P. vulagaris*, *Sesbania sesban*	[67,68,69]
*R. etli*	*P. vulagaris*	[67,68]
*R. leguminosarum*	*Sesbania sesban*, *V. faba*	[69,70]
*M. amorphae*	*S. sesban*, *C. arietinum*	[69,71]
*M. plurifarium*	*S. sesban*	[69]
*R. huautlense*	*S. sesban*, *C. arietinum*	[69,71]
*M. plurifarium*	*S. sesban*, *C. arietinum*	[69,71]
*B. elkanii*	*Vigna subterranea*, *Glycine max*, *A. hypogaea*	[72,73,74]
*B. japonicum*	*V. subterranea*, *Glycine max*, *A. hypogaea*	[72,73]
*M. ciceri*	*Cicer arietinum*	[71]
*M. mediterraneum*	*C. arietinum*	[71,75]
*M. loti*	*C. arietinum*	[71]
*M. opportunistum*	*C. arietinum*	[71]
*M. haukuii*	*C. arietinum*	[71]
*M. tianshanense*	*C. arietinum*	[71]
*M. cicero*	*C. arietinum*	[75]
*B. yuanmingense*	*A. hypogaea*	[73]
*B. canariense*	*A. hypogaea*	[73]
*B. liaoningense*	*A. hypogaea*	[73]
*R. pisi*	*V. faba*	[70]
*R. anhuiense*	*V. faba*	[70]
*R. laguerreae*	*V. faba*	[70]
*R. binae*	*V. faba*	[70]
*R. bangladeshense*	*V. faba*	[70]
*R. lentis*	*V. faba*	[70]
*R. aethiopicum*	*V. faba*	[70]
*R. aegypticum*	*V. faba*	[70]

In the recent past, many studies investigated the diversity of rhizobia in specific African regions. These studies uncovered a similarity in the identity of rhizobia distribution in different areas of the world. The introduction of new legumes in Africa, such as the common beans, which do not have an African origin, might have also introduced foreign symbionts into African soils. In the previous study, rhizobia *R. etli* was recovered from seeds with a damaged seed coat [76]. For instance, *R. etli*, *R. leguminosarum*, *R. tropici*, *R. meliloti*, *Bradyrhizobium japonicum*, *B. elkani*, and *R. phaseoli* were isolated from East African soils and identified with various techniques such as PCR restriction fragment length polymorphism (RFLP) of the 16S rRNA gene or whole-genome sequence analyses [17,66,67]. In west Africa, 58 common bean isolates from Senegal and Gambia were identified as *R. etli* and *R. tropici* [68]. In fact, *R. tropici*, *R. leguminosarum*, *Mesorrhizobium amorphae*, *M. plurifarium*, *R. huautlense*, and *M. plurifarium* dominated Nigerian soils [69].

Furthermore, in Ghanian soils, *B. elkanii* and *B. japonicum* were the dominant isolates from Bambara groundnut and soybean cultivars [72]. Most taxonomical studies are based on 16S rRNA phylogeny. However, the resolution power of the 16S rRNA gene sequences for individual bacterial species is limited. Most of these studies only allow the taxonomical classification of rhizobia up to the genus level; for example, according to a study from South Africa [77], *Bradyrhizobium*, *Ensifer*, *Mesorhizobium*, *Rhizobium*, and *Paraburkholderia* were the most predominant rhizobia genera in acacia nodules. Likewise, the genera *Rhizobium*, *Bradyrhizobium*, *Mesorhizobium*, and *Sinorhizobium* were isolated from nodules of wild legumes in Egyptian soils [78]. These examples demonstrate that the origin of many African rhizobia is not well understood. Consequently, their performance under different conditions can differ substantially.

Moreover, symbiotic rhizobia have been characterized in cowpea (*Vigna unguiculata*), including species of *Rhizobium* and *Bradyrhizobium.* In a study conducted in Botswana, Pule-Meulenberg et al. [79] characterized cowpea rhizobia symbionts, including *B. youanmingense*, *B. japonicum*, and *B elkanii*. In Mozambique, Simbine et al. [2] characterized *B. zhanjaingese*, *B. yuamingese*, *B. cajani*, *B. vignae*, *B. paxallaeri*, *B. icense*, *B. elkanii*, *B. pachyrhizi*, *B. mercantei*, *B. erythrophlei*, and *B. namibiense* in cowpea. In Kenya, Muindi et al. [80] reported the symbiotic efficiency of rhizobial endophytes associated with cowpea in semiarid areas, including *Rhizobium alamii*, *R. mesosinicum*, *R tropici*, and *R. punense*. A related study in Kenya also reported the distribution of *R*. *pusense*, *R*. tropici, and *Mesorhizobium* sp. [81]. Kebede et al. [82] assessed cowpea-nodulating rhizobia in Ethiopia and presumptively categorized them into *Rhizobium* and *Bradyrhizobium* species.

In addition, rhizobia with nitrogen fixation ability have been characterized in chickpea (*C. arietinum* L.). For instance, various *Mesorhizobium* were characterized in Ethiopia, including *M. ciceri* and *M. mediterraneum*, *M. amorphae*, *M. loti*, *M. plurifarium*, *M. opportunistum*, *M. haukuii*, and *M. tianshanense* [71]. In Tunisia, the symbiotic performance of *M. mediterraneum* and *M. cicero* in chickpea was also observed [75]. In peanut (*A. hypogaea*), the most commonly reported rhizobia are *B. japonicum*, *B. elkanii*, *R. giardini*, and *R. tropici* [73,83]. Hassen et al. [84] characterized rhizobia species *B. japonicum*, *B. elkanii*, and *B. yaumingense* as critical forage legume nodulators in South Africa. In the root nodules of lentil (*Lens culinaris*) and faba bean (*V. faba*) collected from southern Ethiopia, *R. pisi*, *R. leguminosarum*, *R. anhuiense*, *R. laguerreae*, *R. binae*, *R. bangladeshense*, *R. lentis*, *R. aethiopicum*, and *R. aegypticum* were identified through phylogenetic analysis of the housekeeping gene *rec*A and symbiotic genes *nod*C and *nif*H [70]. In Mozambiquan soils, *B. elkanii* were found to be the most dominant soya bean-nodulating isolates, and their distribution is markedly influenced by pH [74].

There is an emerging consensus in the literature that *Bradyrhizobium* is a cosmopolitan and diverse bacterial group that nodulates a wide range of host legumes in Africa; however, the diversity and distribution of bradyrhizobial symbionts among diverse nodulating indigenous African legumes are poorly understood, even though increased food legume production is required. Early studies have identified this genus as the dominant microsymbiont for legumes in Senegal, Ethiopia, Mozambique, South Africa, Botswana, and Namibia [85]. However, there is scanty information on legume nodulation by *Bradyrhizobia*.

## 4. Factors Affecting the Distribution of Rhizobia in Africa

According to Puozaa et al. [86], the physicochemical properties of soil serve a critical role in the diversity of microorganisms present in a particular habitat. Environmental factors significantly affecting rhizobia distribution and diversity in Africa include temperature, pH, salinity, drought, and pesticides (Figure 3) [87].

Temperature is a significant factor in legume–rhizobia interactions and determines the strains’ distribution and nodulation ability [88]. For optimal performance, each combination of rhizobia and legumes requires specific conditions. The ideal temperature for rhizobia growth is 25–30 °C, although some can tolerate up to 42 °C in arid and semiarid areas. In some regions of Africa, the temperature can go up to 60 °C. High temperature reduces nitrogen fixation [89] mainly through reduced nitrogenase activity [90,91]. Decreased glutamate synthase and glutamine synthetase activities lower nitrogen assimilation in heat-stressed plants [92]. Genetic modifications such as loss of plasmids [93] and genome rearrangement [94] have been reported for some rhizobia strains exposed to temperature stress. Furthermore, high temperatures reduce the release of *nod* gene induction signals (specific flavonoids) [95] and, consequently, the nodulation process [96] in common beans. Although some rhizobia strains fix nitrogen at temperatures as low as 4 °C [97], nodule formation is severely impaired because fewer flavonoids are secreted [88].

Isolation of native rhizobia species from extreme temperature ranges, such as those found in Africa, is required to obtain high-temperature tolerant rhizobia species [98]. In Egypt, 68 rhizobia species were isolates that could grow at temperatures ranging from 20 °C to 35 °C, although some grew at a maximum of 50 °C [99]. Solomon and Lehman [3] successfully isolated high-temperature tolerant rhizobia from common beans in Tanzania. However, there is still a need for more studies concerning the temperature and its effect on the rhizobia population across different regions in Africa.

Biological nitrogen fixation is effective in neutral or slightly acidic soils [98]. However, most African soils are acidic because of industrial pollution and poor agricultural practices, such as the overutilization of inorganic fertilizers (Figure 4) [8,100]. In general, acidic soils negatively affect crop production [67,101,102] and the proliferation of soil bacteria. When the pH is below 5.5, the soil is considered acidic, and the solubilization of toxic metal ions is promoted, severely increasing metal ion toxicity [88]. Acidic conditions and the presence of heavy metals are detrimental to rhizobia growth, symbiosis, and nitrogen fixation. Reduced rhizobia growth restricts their distribution, propagation, nodulating ability, and survival [89]. In particular, low pH promotes Al toxicity [103], which is more detrimental to rhizobia than acidity or phosphorus and calcium deficiencies [104]. At pH 5.4, Kawaka et al. [8] found Al concentration as high as 2.8 Cmol/kg, and Mulama et al. [100] found 1.2 Cmol/kg in Kenyan soils. In particular, exposing rhizobia to Al over extended periods of time has long-term negative consequences on the rhizobia population and its growth [105].

Al toxicity in rhizobia might be caused by the direct binding of polymeric Al ions to the cell membrane [106]. However, when such polymeric compounds were removed from the media before inoculation, Al toxicity to rhizobia was not eliminated [103]. Another explanation for the toxicity of Al could be that the ions bind to the phosphate moieties of the DNA helix after penetrating the cell membrane. This increases its stability and prevents or slows down its replication [107]. Though significant in rhizobia research, this theory was also questioned because rhizobia, like any other bacteria, maintains its internal pH values slightly alkaline (pH 7.2–7.5) and, therefore, intracellular Al is expected to precipitate out [108]. Interestingly, Al remained toxic to rhizobia even after raising the pH values greater than 6 [103]. Finally, [105] showed that the binding of Al to the cell membrane of rhizobia increased the cell permeability, leading to extracellular leakage of cytoplasmic contents. Rhizobia responds to acid and Al toxicity by increased synthesis and extracellular transport of exopolysaccharides [109]. Furthermore, biofilm formation, membrane repair, stabilization, and biogenesis were previously reported as some of the responses of rhizobia to Al toxicity [105]. There is general agreement that the toxic effects of Al to rhizobia are caused by varying mechanisms, which may also be different among rhizobia strains. These studies highlight the significant problems associated with many African soils’ acidity and Al contaminations.

Soil salinity is another global problem that negatively impacts agricultural productivity and sustainability. Soil salinity occurs in arid and semiarid regions where rainfall is insufficient to meet the water requirements of the crops and microbial populations. In Africa, about 80 × 10^6^ hectares are either saline, sodic, or saline–sodic, of which the Sahel in West Africa is the most affected region [110]. Furthermore, flooding, seepage, over-irrigation, silting, and a rising water table are the main reasons for the increased salinization in Africa [111]. Salinity has harmful effects on both plant hosts and bacteria and restricts the survival of rhizobia, colonization, and nodulation activity. In a recent study, plants growing at 2, 4, 6, and 8 mS/cm salt experienced a mean reduction in nodulation ability between 60 to 93% [112]. Another study showed that at 0.8% NaCl, rhizobia growth was inhibited while nodulation was eliminated in soya beans [113]. Nonetheless, some strains evolved mechanisms to tolerate high salinity levels in the soil [114]. These rhizobia were isolated from wild legumes found in the arid areas of Africa, and they form effective nodules and fix nitrogen in a high-alkalinity environment [78]. Whether those rhizobia are agricultural relevant remains to be determined.

In some African soils, rhizobia growing and maintaining effectiveness in nitrogen fixation in a wide range of pH has recently been isolated. For example, rhizobia with the ability to grow at pH as low as 3.5 and as high as 10 [115] were isolated in Egypt. In South Africa, Botha et al. [116] found that *Bradyrhizobia* population serotype 135 for soybeans was adapted to alkaline soils. In contrast, serotype 122 was adapted to neutral or acid soils and could not thrive in alkaline soils. A commercial strain of *B. japonicum* (strain 532c) was previously shown to boost soybean yield in Ethiopia, Kenya, and Nigeria. However, because of the acidic conditions, it did not work in Ethiopian soils [117].

Because of climate change, large areas of sub-Saharan Africa experienced extended drought conditions in recent years, with severe consequences for plant and microbial growth in the soils [118]. Water stress disturbs the formation of nodules and leghaemoglobin biosynthesis [119] and results in the accumulation of toxic nitrogenous products due to impaired transport of nitrogen fixation products. Exposure of *Lablab purpureus* (L.) roots to mild drought stress moderately reduced its nitrogenase activity and significantly reduced, to about 25–35%, the control treatments under severe stress [120]. The glutamine synthetase is even more sensitive to drought stress than the nitrogenase, and its activity reduces faster than that of the nitrogenase. Currently, little attention is given to research on the association between soil moisture and rhizobia in Africa. The majority of Africa’s arid lowlands have a low moisture content and a wide annual temperature range. As a result, finding good rhizobia candidates for developing effective symbioses in drought conditions useful for the production of common bean and other legumes will almost certainly arise from the successful isolation of rhizobia from such an environment [98].

In 2019, three East African countries, Kenya, Ethiopia, and Somalia, witnessed the worst desert locusts invasion [121]. The governments and the United Nations Food and Agriculture Organization (FAO) embarked on an intensive, coordinated ground and aerial pesticide spraying campaign over eight East African countries to protect the pasture for livestock, especially in semiarid regions, and the food supply for millions of people [121]. However, more than 95% of the pesticides used in managing these pests are toxic to animals and the environment [121]. Pesticides do not seem to affect the microbial population in the soil directly, although concentrations exceeding the recommended amounts impaired microbial activities and induced shifts in their populations [122]. A subset of these agrichemicals, the organochlorine pesticides, contaminated the environment with the accumulation of toxic breakdown products which inhibit or delay the recruitment of rhizobia bacteria to the host plant roots, reduce nodulation, lower the nitrogenase activity, and reduce the overall plant yield [123]. Many organochloride and organophosphate pesticides also reduce the number and diversity of rhizobia in the soil [88], thus directly influencing the soil’s rhizobial community.

**Figure 4 ijms-23-06599-f004:**
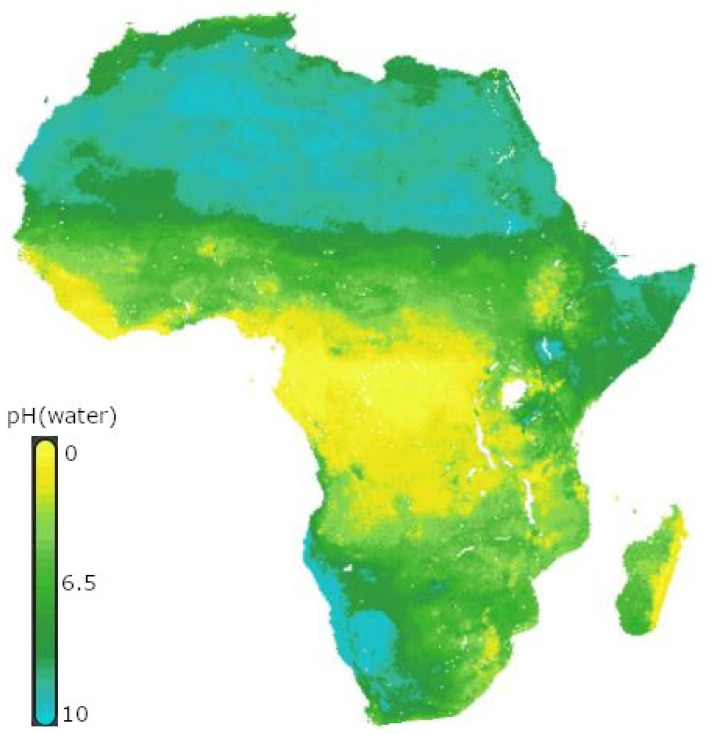
Soil pH in Africa Africa [124].

## 5. Methods Used in Rhizobia Characterization in Africa

According to Koskey et al. [125], efficient methods classify rhizobia according to their genotypes for exploitation. Recently, more efficient molecular techniques complemented traditional morpho-cultural and phenotypic techniques in characterizing microbial strains. Gene and genome sequencing technics, in combinations with electrophoresis and polymerase chain reaction (PCR), allow faster and more efficient distinguishments of even closely related species [126]. Moreover, metagenomics and third-generation gene sequencing methods are more reliable, sensitive, and faster than traditional methods [127,128].

### 5.1. Phenotypic and Biochemical Characterization

Conventional phenotypic identification of rhizobia entails various methods, including observing colony morphology (size, shape, color, texture, and general appearance), growth of the strains in different media, and microscopy [6,125]. However, phenotypic methods have poor precision and cannot account for variable characteristics among members of the same species [129]. Many traditional methods are inconclusive and can only be used for preliminary identification of rhizobia. However, biochemical reactions reveal important information for identifying the genera of the bacteria, and unique enzymatic profiles provide helpful additional information. Most commonly used biochemical tests include Bromothymol Blue (BTB) agar tests [130] to differentiate between acid and alkaline producers [8,17]. In the dark, rhizobia stand out as white, translucent, glistening, elevated, tiny colonies with an entire margin, in contrast to the red-stained colonies of Agrobacterium when grown in the dark [131]. Since all rhizobia are positive for catalase, indole, citrate, and oxidase tests, they are often used to distinguish rhizobia from other bacteria [80,100,132].

### 5.2. Molecular Characterization

Koskey et al. [125] used amplified ribosomal DNA restriction analysis (ARDRA) and PCR to determine the genetic diversity of rhizobia in Eastern Kenya. ARDRA, i.e., amplification and restriction of 16S rDNA sequences, is preferred because it is less expensive, has a high taxonomic resolution, and does not need specialized equipment, such as fingerprinting techniques [133]. Other methods used to evaluate the diversity of rhizobia include amplified fragment length polymorphism (AFLP), restriction fragment length polymorphism (RFLP), random amplified polymorphic DNA (RAPD) [86,134,135], and BOX-PCR, which uses the BOX-A1R primer that targets repetitive extragenic palindromic sequences interspersed throughout the genome [136]. The use of BOX-PCR effectively characterized 130 Kersting’s groundnut rhizobial isolates from contrasting environments in Ghana, South Africa, and Mozambique [137]. Furthermore, highly conserved 16S rRNA and 16S–23S rRNA sequences have been used to determine the taxonomic position of different strains [138]. Occasionally, SDS-PAGE analysis of whole-cell proteins, DNA–DNA hybridization, and multi locus sequence analysis (MLSA) were used [4]. Several universal and rhizobia-specific primers for sequencing and DNA amplification generated rhizobia phylogenies [125]. In addition, 16S–23S rRNA ITS regions were recently used as phylogenetic markers for subspecies delineation [139]. Because of its high sequence variation, the ITS region allows the distinction of even closely related species. For instance, Mwenda et al. [66] investigated rhizobia with high genetic diversity, and 43 ITS genotyping allowed them to be grouped into 10 clusters. Moreover, through genetic DNA fingerprints, 16S rRNA, and PCR-RFLP analyses, 178 isolates were recovered from the root nodules of *P. vulagaris* and classified according to their phylogenetic relationships.

According to Puozaa et al. [86], the most popular tool for bacterial phylogenetic studies is 16S rRNA sequencing because the gene contains a conserved subunit that does not readily undergo mutation [4]. The 16S rRNA sequence from an unknown isolate is compared to reference sequences in the database, usually the NCBI nonredundant database. Relatedness between sequences is inferred from a mathematically computed gene sequence identity (SI), of which 97% is taken as species demarcation value. However, several challenges have been observed in using 16S rRNA sequence in bacterial taxonomy. Its low evolutionary rate reduces its resolving power, especially when comparing recently diverged or closely related species. For example, a comparison of the16S rRNA gene sequences of three psychrophilic strains resulted in 99.5% SI, clearly indicating that they belong to the same species; however, DNA–DNA hybridization results demonstrated that these strains belong to different species [140]. In some bacteria, multiple nonidentical 16S rRNA genes have been reported, e.g., up to 15 copies of 16S rRNA genes per genome with high mean intragenomic diversity were identified in a study comparing 883 bacterial genomes [141]. Homologous recombination and horizontal gene transfer involving 16S rRNA genes have been shown for some bacteria [142], which may lead to misinterpretations.

Considering limitations faced by 16S rRNA taxonomy, species identification by sequence comparison of more than one gene is more reliable. MLSA, a method that collects sequence information of several housekeeping genes, is often used for the phylogenetic analyses of bacteria [143]. For instance, MLSA for the three housekeeping genes *rec*A, *atp*D, and *dna*K allowed better discrimination of rhizobia isolated from root nodules of white lupin (*Lupinus albus* L.) in Tunisian calcareous soils [144]. Nucleotide sequences for *rec*A, *gln*II, *rpo*B, and partial 16S rRNA genes were used to characterize 32 native rhizobia associated with common bean (*P. vulagaris* L.) in Ethiopia [145]. Furthermore, MLSA of three housekeeping genes, *rec*A, *atp*D, and *rpo*B, allowed the identification of 58 rhizobial strains from root nodules of *V. faba* in Eastern Algeria [146].

Since its inception, DNA–DNA hybridization (DDH) has improved bacterial taxonomy [147], especially when their 16S rRNA gene sequences show more than 97% similarities [148]. DDH is based on the fact that the degree of hybridization between two genomes is proportional to their relatedness. The melting temperature of the hybrid genome is compared to that of the single reference genome, and if it is 70% or more, the two strains belong to the same species. Although many researchers in Africa have not used wet-lab DDH, Beukes et al. [149] used it to characterize twelve Paraburkholderia strains isolated from indigenous South African fynbos legume *Hypocalyptus sophoroides*. DDH has several limitations: it is time-consuming, unavailable for nonculturable bacteria, and ill-suited for rapid identification [150]. It has also been reported that results vary between laboratories where the hybridization is performed [4]. Furthermore, the proposed 70% values for species discrimination do not correspond to an evolutionary theory-based concept of what properties a species should have [151].

Alternatives to wet-lab DDH are the whole genome’s overall genome-related indices, such as digital DDH (dDDH) [152] and average nucleotide identity (ANI) values [153] whose computation involves both similarity in gene content and nucleotides of the shared genes. Furthermore, GC content usually analyzed along with DDH can easily be calculated from the whole genome sequences. 70% in dDDH, 95% in ANI [154], and 1% G + C content [155] values are accepted species delimitation boundaries because they coincide with the 70% DDH value. The dDDH is based on genome blast distance phylogeny (GBDP), originally devised as a whole-genome sequence phylogenetic tree inference [156]. It was, however, later evaluated to infer digital equivalents for DDH values [152] which mimic the wet-lab hybridization results. In this method, two genomes are aligned to produce a set of high-scoring segment pairs (HSP). The information contained in these HSPs is then transformed into a single genome-to-genome distance value using a suitable distance formula [155]. ANI is the average nucleotide identity of all orthologous genes shared between two genomes. ANI offers a strong resolution between strains of the same or closely related species. It can be approximated even between the draft genomes recovered from environmental samples in metagenomic studies or single-cell techniques that do not encode universally conserved genes but encode at least a few shared genes [157]. This dramatically expands the number of sequences that can be classified compared with a universal gene-based approach.

In the recent past, the use of ANI and dDDH has gained prominence amongst African researchers. Steenkamp et al. [158] employed dDDH and ANI indices in the taxonomy of twelve South African root-nodulating *Burkholderia* isolates from native papilionoid legumes. Together with MLSA, Diouf et al. [159] identified three new species of *Mesorhizobium* that nodulate with wild Senegales *Acacia senegal* (L). An isolate associated with Côte d’Ivoire’s pigeon pea (*Cajanus cajan*) and previously identified as *B. elkanii* by 16S ribosomal RNA (rRNA) genes phylogeny was later found to be a novel species after utilizing the ANI [160]. Using both ANI and dDDH, Wekesa et al. [17] characterized two high-efficiency Kenyan common bean-nodulating rhizobia. While investigating the taxonomic status of 31 rhizobial isolates from the root nodules of diverse South African legume hosts, a *Papilionoideae* isolate previously identified as *Paraburkholderia tuberum* was found to be a new species and consequently renamed *Paraburkholderia podalyriae* after performing average nucleotide identity [161].

In recent years, it has been realized that not all microbes in the environment are culturable. Consequently, new methods have emerged to characterize the bacteria without culturing them. Metagenomics is a culture-independent approach for gaining access to the genetics and physiology of microorganisms in any habitat [162]. DNA is directly extracted from the environmental sample and used for microbial identification for this method. In shotgun metagenomic sequencing, DNA isolated from a diverse microbial community is broken into a collection of small fragments, followed by independent sequencing of each fragment. This enables a comprehensive sampling of all genes in all organisms present in a complex sample enabling functional analysis of isolates and taxonomy.

## 6. Potential and Challenges to Commercialization of Rhizobia-Based Inoculants

The eradication of food insecurity, poverty, and hunger is a crucial agenda identified in the United Nations Sustainable Development goals (SDGs). About 700 million people do not have access to sufficient food, most of them from Latin America, Asia, and Africa [163]. The cost of commercial fertilizer is way beyond the affordability of poor African farmers. More so, overuse of fertilizer has been identified as a causative agent of soil and water pollution. The ongoing COVID-19 pandemic has exacerbated the situation. Therefore, there is a need to embrace clean and affordable agricultural practices, such as plant growth-promoting microorganisms, to enhance plant productivity with reduced environmental effects.

The use of rhizobia in BNF is a viable strategy to enhance crop yields through nitrogen fixation [164,165] and ensures that the appropriate strains of rhizobia for the specific legume exist in the soil. The popularity of BNF technology has recently improved due to its ability to reduce overreliance on inorganic fertilizers [166,167]. Apart from fixing nitrogen, rhizobia inhibit disease-causing phytopathogens by chelating iron away or production of antibacterial agents in the rhizosphere [168,169], solubilizing phosphate [170], and supplying growth-promoting hormones such as IAA to host plants [17]. Inoculating legumes with rhizobia is the most used technique in BNF [4,171]. Native rhizobia inoculated on different legumes, including peanut, common bean, soybean, cowpea, lentils, and green grams, have improved their productivity [16,18,172,173,174,175].

The selection of elite rhizobia strains as commercial inoculants is based on various considerations. They should effectively compete with native rhizobia for nodule occupancy, and the strains must have a high nitrogen fixation ability for the intended host under greenhouse and field conditions [176,177]. Moreover, they should possess characteristics such as stress tolerance, satisfactory growth, and genetic stability when the inoculum is being manufactured [178]). Moreover, Checcucci et al. [176] noted that the strain should possess limited year-to-year persistence in unplanted soil so that newly improved strains can be introduced without concern about competition. Elite rhizobia selected as seed and soil inoculants in field trials have shown increased crop yields in Africa [13,176]. Other reports [18,179,180] of increased use of microbial inoculants in smallholder farming systems in Africa are encouraging. Therefore, the use of appropriate inoculants in legumes offers an opportunity to improve legumes’ productivity. However, the technology has not been well established in Africa, yet African farmers can immensely benefit from its affordability [181].

A review of the rhizobia-based inoculants revealed that within the African content, few countries had progressed towards the commercialization of the elite rhizobia strains (Table 1). In Africa, South Africa has many rhizobia inoculant products and manufacturers. The leading producers include Soygro (Pty) Ltd. (Potchefstroom, South Africa), Biological Control Products SA (Pty) Ltd. (Emeryville, CA, USA), and Microbial solution Ltd. (Parnell, New Zealand) [182]. These companies manufacture a range of rhizobial inoculants for various crops (Table 2). Perhaps the significant boost for rhizobia inoculum technology came from the Bill and Melinda Gates Foundation in the other African regions. It mandated TSBF-CIAT (Tropical Soil Biology and Fertility-Center for Tropical Agriculture) to scientifically evaluate and select effective commercial products to enhance and sustain crop yields in designated agroecological zones in three African countries [183]. Based on this, research institutions, the private sector, and universities have partnered with farmers in Africa to deliver efficient inoculants. For example, in Kenya, the Microbiological Resources Center Network (MIRCEN) partnered with the University of Nairobi and MEA Fertilizer Ltd. in the early 1980s to develop Biofix, a rhizobia-based bioinoculant for common beans, soya beans, cowpea, and groundnuts [184,185]. Another rhizobia-based inoculant in the Kenya market is KEFRIFIX, produced by the Kenya Forestry Research Institute (KEFRI) [186]. In Tanzania, the Sokoine University of Agriculture, in collaboration with the FAO, the Kenyan Ministry of Agriculture, and some nongovernmental organizations, developed NITROSUA as a soybean and lucerne inoculant [187,188]. BIO-N FIX is a commercial soybean inoculant containing about 10^9^ viable cells of *B. japonicum* strain USDA110 with origin from the University of Hawaii NifTAL Project produced by Makerere University with the aid of the United State Agency for International Development (USAID) [189]). In addition, there is ongoing research to commercialize more N-fixing bioinoculants [190,191]). The IITA (International Institute of Tropical Agriculture) collaborating with Nigeria through its business incubation program has led to the commercialization of Nodumax^®^ (Woomer et al., 2013), which was reported to improve soybean productivity [192]. The N2Africa project is a research and development partnership program aimed at developing, disseminating, and promoting appropriate nitrogen fixation technologies for smallholder farmers [193]. The other countries making some milestones in the manufacture of inoculants are Rwanda and Zambia, though at a small scale [189].

**Table 2 ijms-23-06599-t002:** Status of the commercialization of rhizobia-based inoculants in Africa.

No.	Name of Inoculant	Rhizobia Strain	Manufacturer, Country	Target Crops	Reference(s)
1	Biofix	Various strains	MEA Ltd., Kenya	Common bean, soybean, lucerne, peas, cowpea, groundnuts	[185,194]
2	Nodumax	*Bradyrhizobia*	IITA, Nigeria	Soybean	[195]
3	Sojapak	*B. japonicum*	Soygro, South Africa	Soybean	[13]
4	Soyflo	*B. japonicum*	Soygro, South Africa	Soybean	[196]
5	Peanutflo	*Bradyrhizobium* sp.	Soygro, South Africa	Peanut and Bambara nuts	[196]
6	Rhizoflo	*Bradyrhizobium japonicum*	BASF South Africa (Pty) Ltd.	Soybean	[186]
7	SeedQuestR	*Rhizobium*	Soygro, South Africa	Soybean	[197]
8	Organo	*Rhizobium*	Microbial solution Ltd., South Africa	Soybean	[197]
9	Likuiq Semia	*Bradyrhizobium elkanii*	Microbial Solution Ltd., South Africa	Soybean	[198]
10	Histick	*B. japonicum*	BASF South Africa Ltd., South Africa	Soybean	[197]
11	Organico	*Rhizobium*	Amka Products	Soybean	[197]
12	Nitrasec Alfalfa (Lucerne)	*Sinorhizobium meliloti*	Microbial solution (Pty) Ltd., South Africa	Lucerne	[199]
13	RAB inoculant	*Rhizobium tropici*	Rwanda Agricultural Board, Rwanda	Common bean	[187]
14	N-Soy	*B. japonicum*	BioControl Products SA (Pty) Ltd., South Africa	Soybean	[197]
15	N-Bean	*Rhizobium phaseolus*	BioControl Products SA (Pty) Ltd., South Africa	Common bean	[197]
16	Nitrosua	*B. japonicum*	Sokoine University of Agriculture, Tanzania	Soybean	[187,189]
17	Nitrozam	Various rhizobia strains	Mt. Makulu Research Station, Zambia	soybean, lucerne and common bean	[189]
18	Bio-N fix	*B. japonicum* and *Rhizobium tropici*	Madhavani Ltd. and Makerere University, Uganda	Soybean and common bean	[187]
19	Kefrifix	Rhizobium	Kenya Forestry Research Institute (KEFRI), Kenya	Leguminous trees, common bean	[186]

Compared with the global biofertilizer market in countries such as the United States, Canada, Argentina, Europe, China, and India, the potential benefits of biofertilizers have been largely untapped in Africa due to a myriad of challenges. One of the challenges facing the commercialization of rhizobia for BNF is the effect of indigenous rhizobia and other bacteria in the soil. It has been noted that when other factors are constant, lower indigenous rhizobia levels lead to a higher response of the inoculums. At the same time, it becomes difficult to improve nitrogen fixation if native populations exceed 10^2^ cells per gram of soil [4]. Therefore, it is important first to determine the rhizobia population in the soil, followed by their characterization.

Another critical challenge is inefficient production technology for commercial strains [178,200] tailored to the local conditions, especially product shelf life, given the high temperature that is unsuitable for prolonged strain survival in some African regions, and this may lead to substandard/poor quality inoculants found in a number of countries [201,202]. In this regard, instead of including only foreign isolates in the inoculants, studies should be performed to identify elite isolates from African soils with the capacity to withstand these harsh conditions.

The other challenges identified include farmer unawareness of the inoculants [203]. A study by Woomer et al. [195] revealed that lack of awareness of the inoculant availability and use is the main limitation to adoption by the farmers. Other studies carried out on the continent show that a knowledge gap between the researchers, extension officers, and farmers regarding the benefits of inoculants [132,204] negatively affected the awareness of farmers on the technology. Therefore, the lack of awareness leads to limited demand for the products and dwindling the production volumes. Other challenges reported hindering the full exploitation and commercialization of the elite rhizobia strains are less developed distribution networks, lack of human capacity, African countries’ inadequate standards and regulatory framework, and lack of private sector involvement [201,203,205]. Furthermore, a study performed by Lesueur et al. [183] identified the commercialization of inoculants with a high level of contaminations as a reason for commercial inoculants’ failure to induce plant growth promotion.

## 7. Future Prospects

Today, it has become increasingly necessary to ensure the sustainability of agriculture due to challenges such as ever-increasing population growth and climate change. In this regard, rhizobial bioinoculants will greatly benefit next-generation agriculture, primarily by utilizing elite strains that combine competitiveness and effectiveness in field conditions. Advances in sequencing and imaging technologies have improved knowledge about the symbiotic process at a biochemical and molecular level. It is expected that eventually, these advances will lead to the development of cost-efficient designs and inoculant products with superior quality readily available to poor African farmers at an affordable cost. Integrated Soil Fertility Management (ISFM) is critical for sustainable intensification in Africa, and legume–rhizobia interaction will have a significant role.

## 8. Conclusions

Enhancing the production of grain legumes is a vital component of agricultural intensification in Africa. The natural ability of rhizobia to convert atmospheric dinitrogen into fixed form for plant use is a costless process that legumes can benefit from, as well as smallholder farmers with limited resources. We have reviewed the factors that impact the distribution and commercialization of elite strains of rhizobia, methods of characterization, opportunities, and challenges facing the commercialization of elite rhizobia in Africa.

## Figures and Tables

**Figure 1 ijms-23-06599-f001:**
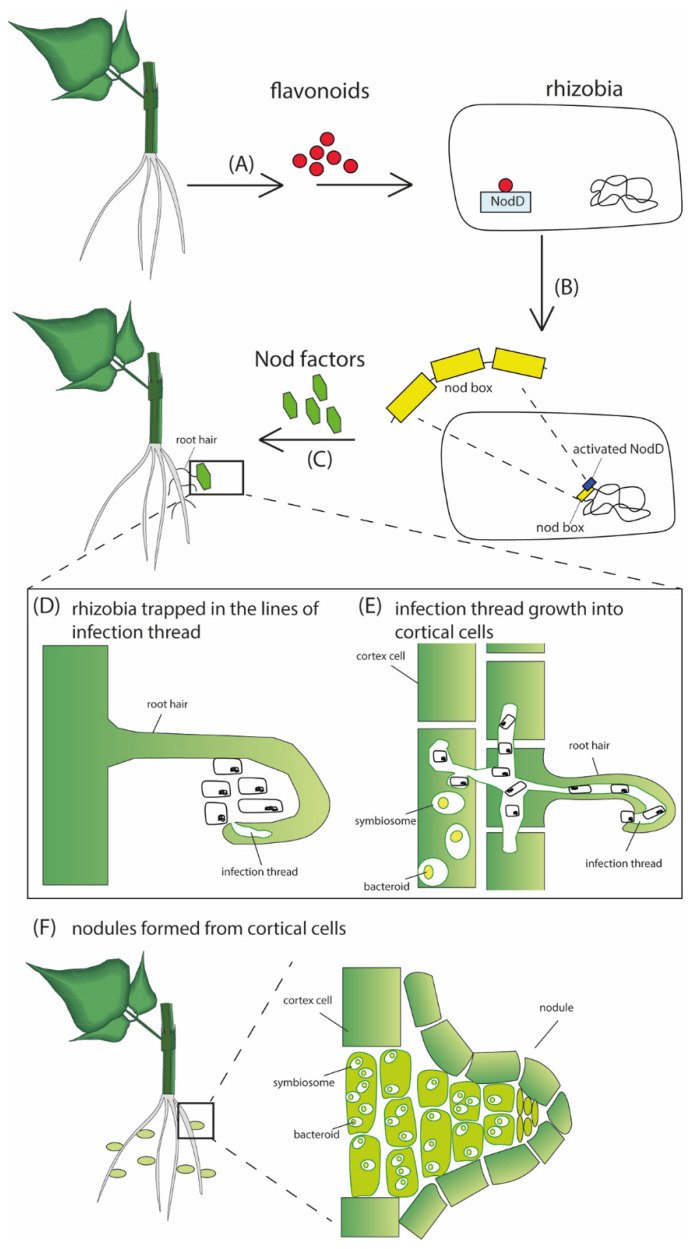
A schematic representation of the nodulation process. (**A**) Host plant releases flavonoids into the rhizosphere that are perceived by specific rhizobia. The flavonoids induce transcription of the genes for biosynthesis of the rhizobial Nod factors, which the plant perceives to allow symbiotic infection of the root. After transcription, the activated NodD binds to the *nod box* promoter (**B**), inducing the transcription and synthesis of Nod factors (**C**). The Nod factors induce the development of the infection thread that traps rhizobia within the curled surfaces (**D**). The infection thread grows through epidermal cells into the cortical cells, where rhizobia are released and internalized by the cortical cells (**E**). Further proliferation and differentiation of both bacteria and infected cortical cells results in nodule formation (**F**).

**Figure 2 ijms-23-06599-f002:**
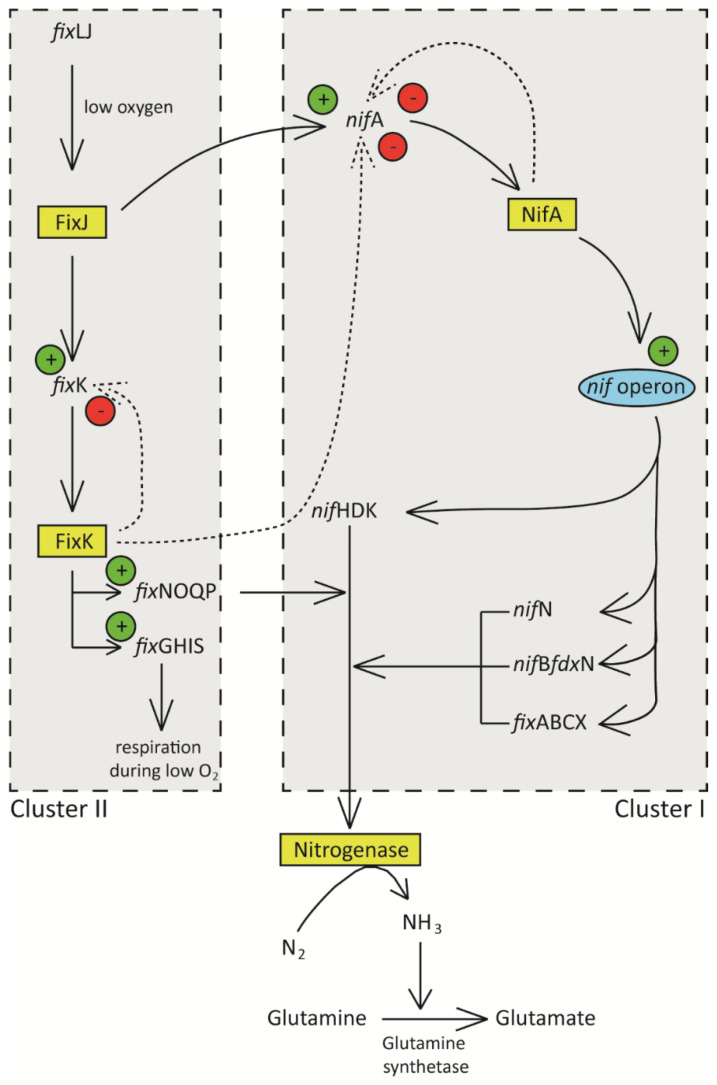
Schematic diagram showing regulation of nitrogen fixation in rhizobia. Low oxygen concentration in the nodules induces transcription of *fix*J, whose product induces the expression of *nif*A and *fix*K. NifA activates the *nif* operon resulting in the expression of *nif* genes that ultimately leads to the synthesis of nitrogenase enzyme and its accessory components. FixK induces the expression of *fix* operon whose products are involved in redox reactions necessary for the nitrogen fixation process. High FixK and NifA result in repression of *nif* and *fix* operons.

**Figure 3 ijms-23-06599-f003:**
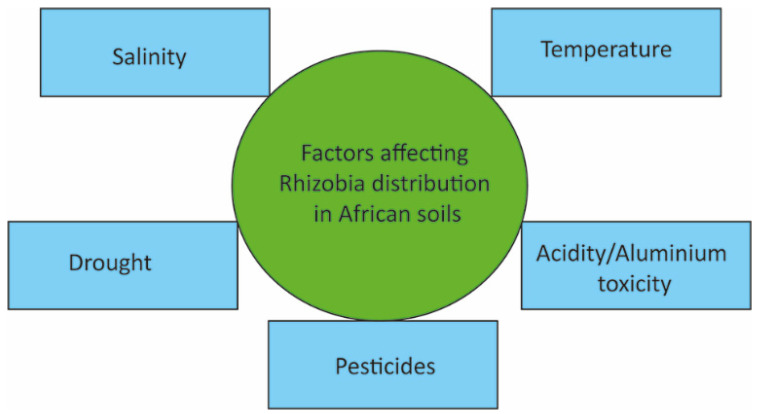
Factors affecting the distribution of rhizobia in African soils.

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
