# Peer review of "Distribution, Characterization and the Commercialization of Elite Rhizobia Strains in Africa"

_ijms, 2022, doi:10.3390/ijms23126599_

Round 1

Reviewer 1 Report

This paper is a review on basic and applied studies on legume rhizobia in Africa. The topic is of great interest for a continent that is needed of efficient, cost-less and environment-friendly agri-technologies to improve food production.

However, this review is yet partly out of focus and certainly incomplete. 

For instance, section 2. Nodulation and nitrogen fixation has been focused on alfalfa rhizobia, being alfalfa not cited as a crop of interest in Africa. I think this section should not focus on a particular symbiosis, but should include more general but key features of legume-rhizobia symbioses: host-bacteria specificity, signal exchange between partners, nodule formation and development, bacteroid differentiation, nitrogen fixation components and regulation, as well as nutrient exchange between plant and bacteroids, and nodule metabolism.   For instance, nothing has been mentioned about leghemoglobin, which is probably the most important symbiotic molecule; virtually nothing about what is the bacteroid or which are the nitrogenase oxygen protection mechanisms in the symbiosis; nothing is mentioned about how the plant regulates nitrogen fixation thru provision of nutrients to the bacteroids, etc., etc.

Regarding section 3, Distribution of rhizobia in African soils, only a few examples of rhizobia nodulating soybeans, common beans and groundnuts are provided. However, there are many more studies on diversity of rhizobia associated with many other grain legumes, such as cowpeas, peanuts, chickpeas, etc, as well as forage legumes and tree legumes, which are missing. Examples of many these symbioses should be also included in this review.

About section 4, factors affecting rhizobia in Africa: this seems like a compendium of main abiotic factors affecting soil fertility and productivity worldwide. Few examples are focused in Africa, with general descriptions of effects of stressful conditions on rhizobial survival and performance. This is particularly true for the pH/Al toxicity and salinity stresses, which cite just a few papers on these topics, with little or none examples in Africa.  Very intriguing is the example of Alnus incana response to drought, since Alnus belongs to family Betulaceae, therefore is not a legume and is not nodulated by rhizobia.

The general descriptions found in previous sections are more evident in section 5, Methods used in rhizobia characterization. Here, classic and modern methods are described, which however are general methods for bacterial diversity and taxonomy studies.

Section 6 is probably the most interesting section of this review, describing several commercial or potentially commercial inoculants in Africa, as well as potential situations limiting the efficacy of these inoculants. Nevertheless, these seem to me too little examples of commercial inoculants in the continent.

In summary, I think this review is of interest, however it needs significantly more focus to the particular but diverse situations in Africa, as well as more extensive compilations of the literature regarding studies of legume nitrogen fixation and rhizobia in Africa.

Author Response

For instance, section 2. Nodulation and nitrogen fixation has been focused on alfalfa rhizobia, being alfalfa not cited as a crop of interest in Africa. I think this section should not focus on a particular symbiosis, but should include more general but key features of legume-rhizobia symbioses: host-bacteria specificity, signal exchange between partners, nodule formation and development, bacteroid differentiation, nitrogen fixation components and regulation, as well as nutrient exchange between plant and bacteroids, and nodule metabolism. For instance, nothing has been mentioned about leghemoglobin, which is probably the most important symbiotic molecule; virtually nothing about what is the bacteroid or which are the nitrogenase oxygen protection mechanisms in the symbiosis; nothing is mentioned about how the plant regulates nitrogen fixation thru provision of nutrients to the bacteroids, etc., etc.

Improved and widen the description. However, we are still willing to do more if the reviewer is not satisfied with our corrections

Regarding section 3, Distribution of rhizobia in African soils, only a few examples of rhizobia nodulating soybeans, common beans and groundnuts are provided. However, there are many more studies on diversity of rhizobia associated with many other grain legumes, such as cowpeas, peanuts, chickpeas, etc, as well as forage legumes and tree legumes, which are missing. Examples of many these symbioses should be also included in this review.

More details and examples included

 About section 4, factors affecting rhizobia in Africa: this seems like a compendium of main abiotic factors affecting soil fertility and productivity worldwide. Few examples are focused in Africa, with general descriptions of effects of stressful conditions on rhizobial survival and performance. This is particularly true for the pH/Al toxicity and salinity stresses, which cite just a few papers on these topics, with little or none examples in Africa. Very intriguing is the example of Alnus incana response to drought, since Alnus belongs to family Betulaceae, therefore is not a legume and is not nodulated by rhizobia.

More details and examples applicable to African studies included and unnecessary studies removed

 The general descriptions found in previous sections are more evident in section 5, Methods used in rhizobia characterization. Here, classic and modern methods are described, which however are general methods for bacterial diversity and taxonomy studies.

Changed and more examples from where these methods were used in Africa added and those without use in rhizobia taxonomy removed

 Section 6 is probably the most interesting section of this review, describing several commercial or potentially commercial inoculants in Africa, as well as potential situations limiting the efficacy of these inoculants. Nevertheless, these seem to me too little examples of commercial inoculants in the continent.

Detailed information and examples added.

Reviewer 2 Report

My comments and Suggestions for Authors are included in the attached File.

Author Response

  1. INTRODUCTION

Sentence lines 36-37: “Diazotrophs in the genus Rhizobium belong to Alphaproteobacteria and Betaproteobacteria”. This is not true. You can say that rhizobia bacteria (in general) belong to Alphaproteobacteria, Betaproteobacteria (and even Gammaproteobacteria), but bacteria of the genus Rhizobium are Alphaproteobacteria. Please, correct in the manuscript.

Corrected

  1. NODULATION AND NITROGEN FIXATION PROCESSES

In the first paragraph (lines 69-77), the description of the process of infection and nodulation is very deficient, and must be corrected and improved.

Improved the description. However, we are still willing to do more if the reviewer is not satisfied with our corrections

In line 74, a corrected sentence is “This causes the SYNTHESIS AND reléase of nod factors…”.

Corrected

… more events about bacteria infection, proliferation and nodulation must be mentioned

Corrected

Fig 1 also lacks information about nodulation process

Improved

Authors explain studies on R. meliloti. But this name is not correct. This species is no longer named like that. Years ago, its name changed to Sinorhizobium meliloti, and at present it is named Ensifer meliloti.

Removed

Sentence in lines 112: “In rhizobia, the regulation of nitrogen fixation is mainly controled by the amount of oxygen in the nodule, UNLIKE FREE LIVING BACTERIA, WHOSE NITROGEN FIXATION IS REGULATED BY THE AMOUNT OF FIXED NITROGEN IN THE NODULE”. I do not understand this sentence. Authors talk about free-living bacteria, …and their nitrogen fixation in the nodule?? Please explain and correct.

Corrected

… Authors describe only one type of infection of legumes by rhizobia, that involving Ensifer meliloti and infection threads. But, taking into account that this Review is about diversity of rhizobial strains in Africa, it would be more appropiate to describe the different types of infection and nodulation performed by different types of African rhizobial strains in legumes. For example, Bradyrhizobium performs infection by crack entry, etc. What different types of infection and nodules (indeterminate, determinate) develop legumes in Africa in symbiosis with rhizobia? I think this Review would be improved with this information.

Included the reviewer’s recommendations

  1. DISTRIBUTION OF RHIZOBIA IN AFRICAN SOILS

Regarding this Section, I miss a Figure or scheme about the systematics and diversity of rhizobial strains identified in Africa. It may be a Figure of phylogenetic tree(s) including identified rhizobia in Africa, or maybe an Africa map with localized rhizobia-legume interactions in the different Countries, for example. Or other Figure resuming rhizobia or rhizobia-legume diversity in Africa. A Figure of this type will enrich the Review.

Unfortunately, we could not get/think of a figure that will give a distinction of African rhizobia. We find that same isolates appear in all regions of Africa. They may be genetically different but there is no why we could tell that from their names. Again, it is not possible to have phylogenetic tree because we are not working with their sequences.

Line 143: “M. amorphae”. Taking into account that this is the first time that this genus is mentioned, please write instead: “Mesorhizobium amorphae”.

Corrected

Lines 154-155: “Consequently, their performance under the new conditions can differ substantially”. This sentence is no clear. What Authors mean with “new conditions”? Please, explain.

Changed

  1. FACTORS AFFECTING THE DISTRIBUTION OF RHIZOBIA IN AFRICA

Figure with a map of distribution of pH of soils in Africa

Included

Lines 170-171: “Futhermore, high temperatures reduce the release of nod gene induction signals”. Please, explain about “nod gene induction signals”. For me, it is not clear what type of signals the Authors are referring to.

Inserted the word ‘flavonoids’ after signal in brackets

Lines 223-225: This example is about symbiosis Alnus incana-Frankia. This is not a symbiosis involving rhizobia. And this work (Ref 83) is not an African case. So I do not understand why Authors mention this work. There are other published works about reduced nitrogenase activity by drought stress in legume-rhizobia symbiosis, and it would be better an example in Africa.

Changed

Line 211: re-order: first, “survival of rhizobia”; second, “colonization”, third, “nodulation activity”.

Changed

Lines 238-242: change order, put first that pesticides reduce the number and diversity of rhizobia in soils; and second, inhibit or delay the recruitment of rhizobia bacteria to the host plant roots, reduce nodulation, lower nitrogenase activity and reduce the overall plant yield.

Changed

  1. METHODS USED IN RHIZOBIA CHARACTERIZATION IN AFRICA

The description of different methods for rhizobia characterization is interesting and well explained. But few examples for characterization of African rhizobial strains are mentioned or referred. Please, re-write this Section, keeping descriptions of the different methods available, but explaining or mentioning the techniques used for published works on rhizobial characterization in Africa. And which techniques have not been applied yet in studies in Africa.

Changed and more examples from where these methods were used in Africa added and those without use in rhizobia taxonomy removed

Line 270: “amplified fragment length polymorphism (AFLP)”.

Added

  1. POTENTIAL AND CHALLENGES TO COMMERCIALIZATION OF RHIZOBIA-BASED INOCULANTS

I think a more extensive or detailed content in this Section will improve the quality and interest of this Review.

Detailed information and examples added

REFERENCES

Please, check Journal Reference style, because, in the manuscript, different References are in different styles (full Journal name in some cases, abbreviated Journal names in others).

We have been using MDPI reference style plugin of endnote. They have recommended its use in referencing. I will ask them if I should insert references manually.

Reviewer 3 Report

This paper reviews the state of the art of the rhizobial inoculation process in Africa. The rhizobial inoculation is an extremely important procedure that paves the road of the future agriculture, and for that reason it is a permanent trending topic in science. This makes this contribution interesting to be published in this journal and even more so considering the small amount of information available in comparison with rhizobial biodiversity in Asiatic or American soils. The paper is therefore well conceived, and discuss interesting information as distribution, characterization, and commercialization of inoculants, however not all the sections are treated in the same depth reducing the general impact of the whole paper. If the authors correct or complete the information with the suggestions of this referee, I will recommend the publication of this review.

Major points.

Particularly, the section 3 lacks important information that in opinion of this referee should be extended and highlighted in comparison with the rest of the paper. The strain distribution must be revised and completed including (i. e.) a map showing the origin of isolates, a table containing bacteria isolated from its host in a determined region including the reference, and/or a phylogenetic tree of the isolates of distinct African areas. Maybe, it could be interesting to bind the distribution with the factors affecting this distribution (sections 3 and 4) to give more importance to the first word of the review title. This will make the paper much interesting for readers, since the characterization is more or less the same in every paper of this nature. Please, see below several references missed, that should be included containing examples of my comments above,

Concerning the methods used in rhizobia characterization in Africa, I should comment that some information is extracted from papers from locations distinct to Africa as Greece or Brazil (103-107) while some information of other genes used in MLSA has been well stablished in African soils (see Gunnabo et al 2019, 2021), and this should be included in the text. As I mentioned this section tends to be homogeneous in this kind of papers around the world, so if the authors want to stablish a distinction in African soils, maybe they can consider highlighting the particularity of the techniques used in distinct African labs as they briefly mention in lines 267-269.

Minor points.

Line 30: …component of the primary….

Line 38: …plants that fix nitrogen…

Line 74: Nod D induces the expression of more nod genes, not only nod ABCFE. Actually, Nod D induces the expression of all genes or operons preceded by a nodbox.

Line 82: Inducing the transcription of nod genes and therefore the synthesis of nod factors

Line 87: R. meliloti italics

Lines 88-89: Complicate sentence, please rephrase avoiding so many fixations!

Line 96:  Nif S is not mentioned in lines 89-90. Is not located with the other genes?

Lines 111-112: This sentence is weird, as it appears the free-living bacteria regulates nitrogen fixation by the nitrogen in the nodule, maybe you mean combined nitrogen in the soil?

Line 261: There are rhizobial strains that do bind congo red and others that do not.

Page 9 of 17: Different letter sizes or types in lines 290, 298, 307, 308, 324 to 328.

Line 451: Authors have mentioned it previously and it is really important to highlight in the conclusions or future prospects that the future of the inoculations will be based in the use of LOCAL elite strains to avoid the use of commercial strains not adapted to every soil condition, and more important, to promote the local trade

Papers suggested (among others…)

Phylogeny and Phylogeography of Rhizobial Symbionts Nodulating Legumes of the Tribe Genisteae.

Stępkowski T, Banasiewicz J, Granada CE, Andrews M, Passaglia LMP. Genes (Basel). 2018 Mar 14;9(3):163. doi: 10.3390/genes9030163.

Genetic interaction studies reveal superior performance of Rhizobium tropiciCIAT899 on a range of diverse East African common bean (Phaseolus vulgaris L.) genotypes.

Gunnabo AH, Geurts R, Wolde-meskel E et al. Appl Environ Microbiol. 2019;85:1–19

Phylogeographic distribution of rhizobia nodulating common bean (Phaseolus vulgaris L.) in Ethiopia.

Hailu Gunnabo A, Geurts R, Wolde-Meskel E, Degefu T, Giller KE, van Heerwaarden J. FEMS Microbiol Ecol. 2021 Apr 1;97(4):fiab046. doi: 10.1093/femsec/fiab046.

Genetically diverse lentil- and faba bean-nodulating rhizobia are present in soils across Central and Southern Ethiopia.

Asfaw B, Aserse AA, Asefa F, Yli-Halla M, Lindström K.

Insights into the Phylogeny, Nodule Function, and Biogeographic Distribution of Microsymbionts Nodulating the Orphan Kersting's Groundnut [Macrotyloma geocarpum (Harms) Marechal & Baudet] in African Soils.

Mohammed M, Jaiswal SK, Dakora FD. Appl Environ Microbiol. 2019 May 16;85(11):e00342-19. doi: 10.1128/AEM.00342-19. Print 2019 Jun 1.

Identification and distribution of microsymbionts associated with soybean nodulation in Mozambican soils.

Gyogluu C, Jaiswal SK, Kyei-Boahen S, Dakora FD. Syst Appl Microbiol. 2018 Sep;41(5):506-515. doi: 10.1016/j.syapm.2018.05.003. Epub 2018 May 20.

Author Response

 Particularly, the section 3 lacks important information that in opinion of this referee should be extended and highlighted in comparison with the rest of the paper. The strain distribution must be revised and completed including (i. e.) a map showing the origin of isolates, a table containing bacteria isolated from its host in a determined region including the reference, and/or a phylogenetic tree of the isolates of distinct African areas. Maybe, it could be interesting to bind the distribution with the factors affecting this distribution (sections 3 and 4) to give more importance to the first word of the review title. This will make the paper much interesting for readers, since the characterization is more or less the same in every paper of this nature. Please, see below several references missed, that should be included containing examples of my comments above

We could not generate a map showing origin of the isolates because we are not sure if any has the origin in Africa. Most of them we understand that were introduced to Africa together with legumes from the centers of domestigation.

We generated the table (table 1) as the reviewer had recommended.

However, we could not generate a phylogenetic tree showing the isolates because we will need DNA sequences to do that. However, we do not have them. If there is another way to do that, we request the reviewer to propose to us how to do it and we will be very glad.

We also did not patch the isolates to factors affecting their distribution because generally such information is limited in literature. For example, researchers discusses distribution without factors affecting them or factors affecting survival with only model isolates.

We are very grateful to the suggested refferences, we found them very helpful

 Concerning the methods used in rhizobia characterization in Africa, I should comment that some information is extracted from papers from locations distinct to Africa as Greece or Brazil (103-107) while some information of other genes used in MLSA has been well stablished in African soils (see Gunnabo et al 2019, 2021), and this should be included in the text. As I mentioned this section tends to be homogeneous in this kind of papers around the world, so if the authors want to stablish a distinction in African soils, maybe they can consider highlighting the particularity of the techniques used in distinct African labs as they briefly mention in lines 267-269.

We added examples on how various researchers employed particular methods in the studies in Africa and removed those that are not particularly practised in Africa or are not used in rhizobia taxonomy.

Line 30: …component of the primary….

Corrected

Line 38: …plants that fix nitrogen…

Corrected

Line 74: Nod D induces the expression of more nod genes, not only nod ABCFE. Actually, Nod D induces the expression of all genes or operons preceded by a nodbox.

Changed

Line 82: Inducing the transcription of nod genes and therefore the synthesis of nod factors

Changed

Line 87: R. meliloti italics

Removed

Lines 88-89: Complicate sentence, please rephrase avoiding so many fixations!

Changed

Line 96: Nif S is not mentioned in lines 89-90. Is not located with the other genes?

Included nifS on the list of cluster 1

Lines 111-112: This sentence is weird, as it appears the free-living bacteria regulates nitrogen fixation by the nitrogen in the nodule, maybe you mean combined nitrogen in the soil?

Corrected

Line 261: There are rhizobial strains that do bind congo red and others that do not.

Removed

Page 9 of 17: Different letter sizes or types in lines 290, 298, 307, 308, 324 to 328.

Corrected .

Round 2

Reviewer 1 Report

The Ms has been greatly improved with the changes the new additions.

Author Response

There is no comment to respond to. We are grateful to the reviewer for improving our ms.

Reviewer 2 Report

I have found this manuscript much improved over the previous version. Authors have incorporated most of the corrections and suggestions made by the reviewers.

However, several minor points remain to be corrected. I send In the attached file the last version of the manuscript with all my suggestions and corrections. Besides, the format of the References remains uncorrected (some Refs with full name, others with abbreviated name, DOI lack, all this was commented for the previous version).

Author Response

However, several minor points remain to be corrected. I send In the attached file the last version of the manuscript with all my suggestions and corrections. Besides, the format of the References remains uncorrected (some Refs with full name, others with abbreviated name, DOI lack, all this was commented for the previous version).

We have corrected the references according to MDPI requirements. DOI is not part of the requirements as shown here:

https://www.mdpi.com/journal/ijms/instructions

After going through the attached pdf of the previous manuscript, we found no comments. Some text is highlighted in yellow, but we cannot tell what to do with it because there are no comments, or at least we can't see them at our end here.

Reviewer 3 Report

The paper has been improved and I recomend it for its publication in the present format.

But prior to it, two important items have to be changed.

Lines 72-73: Under severe nitrogen starvation, legume roots release flavonoid-containing exudates in the rhizosphere, interacting with nodulation protein D (NodD), which is an N-acetylglucosamine in rhizobia...

NodD is not an N-acetylglucosamine... please revise if you have a copy-paste mistake.... Its a protein that regulates LCO synthesis...

Lines 596-599. in this sectionn you are referring to Table2

Author Response

Lines 72-73: Under severe nitrogen starvation, legume roots release flavonoid-containing exudates in the rhizosphere, interacting with nodulation protein D (NodD), which is an N-acetylglucosamine in rhizobia...

NodD is not an N-acetylglucosamine... please revise if you have a copy-paste mistake.... Its a protein that regulates LCO synthesis...

we have corrected this

Lines 596-599. in this sectionn you are referring to Table2

No, table 2 is a summery of companies and institutions involved in formulating commercial inoculants while lines 596-599 are referring to factors that hinder expansion and growth of African commercial inoculants. However, we have inserted in the text what Table 2 is referring to.